# Fatigue Assessment of Inconel 625 Produced by Directed Energy Deposition from Miniaturized Specimens

**Felipe Klein Fiorentin** [1,2,*], **Duarte Maciel** [1,2], **Jorge Gil** [1,2], **Miguel Figueiredo** [2], **Filippo Berto** [3] **and Abílio de Jesus** [1,2]

1 Institute of Science and Innovation in Mechanical and Industrial Engineering (INEGI), FEUP Campus, Dr. Roberto Frias Street, n 400, 4200-465 Porto, Portugal; up201605880@up.pt (D.M.); jgil@inegi.up.pt (J.G.); ajesus@fe.up.pt (A.d.J.)

2 Mechanical Engineering Department, Faculty of Engineering, University of Porto (FEUP), Dr. Roberto Frias Street, 4200-465 Porto, Portugal; mfiguei@fe.up.pt

3 Department of Mechanical and Industrial Engineering, Norwegian University of Science and Technology, 7491 Trondheim, Norway; filippo.berto@ntnu.no

\* Correspondence: ffiorentin@inegi.up.pt

**Abstract:** In recent years, the industrial application of Inconel 625 has grown significantly. This material is a nickel-base alloy, which is well known for its chemical resistance and mechanical properties, especially in high-temperature environments. The fatigue performance of parts produced via Metallic Additive Manufacturing (MAM) heavily rely on their manufacturing parameters. Therefore, it is important to characterize the properties of alloys produced by a given set of parameters. The present work proposes a methodology for characterization of the mechanical properties of MAM parts, including the material production parametrization by Laser Directed Energy Deposition (DED). The methodology consists of the testing of miniaturized specimens, after their production in DED, supported by a numerical model developed and validated by experimental data for stress calculation. An extensive mechanical characterization, with emphasis on high-cycle fatigue, of Inconel 625 produced via DED is herein discussed. The results obtained using miniaturized specimens were in good agreement with standard-sized specimens, therefore validating the applied methodology even in the case of some plastic effects. Regarding the high-cycle fatigue properties, the samples produced via DED presented good fatigue performance, comparable with other competing Metallic Additive Manufactured (MAMed) and conventionally manufactured materials.

**Keywords:** fatigue; Metallic Additive Manufacturing; directed energy deposition; Inconel 625; miniature specimen

## 1. Introduction

Additive Manufacturing (AM) can be defined as the process of adding material, layer-by-layer, in order to create a new part or improve an existing one [1]. Additive methods differ from conventional subtractive ones, where the material is removed from a workpiece [2], leading to unavoidable material waste [3]. Additionally, AM process does not require design-specific tools, like in injection moulding or casting [4,5]. Regarding the economic viability, AM has found its optimum field on small and medium series production, in contrast with the economy of scale achieved with processes such as casting and forging [6].

Recently, MAM has been consolidated in the manufacturing market. A few years ago, the process was often called rapid prototyping, meaning that it was mainly used to fabricate models and not final functional parts. However, lately thanks to this technological advancement, the MAM became present in the manufacturing process of final components. The presence of MAM in the production of structural functional parts leads to a new

challenge, consisting in ensuring the mechanical properties of these components, i.e., their certification for the target application.

The two most used MAM technologies are Powder Bed Fusion (PBF) and DED, the first one is able to produce more complex parts, but has lower deposition rates (inferior productivity) and the maximum size of parts is generally small [7]. Additionally, at the current state of PBF technology, microstructural defects (pores, for example) are an intrinsic characteristic of the process [6].

MAM microstructures can be very different from those resulting from conventional counterpart methods [8]. Regarding nickel superalloys, often the presence of dendrites oriented in the build direction can be found in AM parts [9]. The MAM of nickel superalloys has been quickly improving in the last years, and, under the right conditions, it is already possible to produce nearly "defect-free" parts [10].

It is essential to determine the mechanical properties of MAM parts. Regarding static mechanical properties, Nguejio et al. [11] had performed extensive research about the tensile properties of Inconel 625 specimens, comparing materials obtained via conventional, PBF and DED processes encompassing the effect of heat-treatment on the tensile properties.

In addition, since several of these parts are subjected to cyclic loads, the determination of fatigue properties are also imperative. Kim et al. [12] have performed a comparison between the fatigue behaviour of PBF and wrought Inconel 625 specimens in a hot environment, and found superior properties of the additively manufactured (AMed) material for a smaller number of cycles to failure (low-cycle fatigue). The high-cycle fatigue of conventional Inconel 625 at room temperature was also studied by Pereira et al. [13], where the fracture surfaces were carefully examined, and a S-N curve was constructed.

Yet, related to the fatigue behaviour of AMed nickel superalloys Nicoletto and his team performed extensive characterization of Inconel 718 obtained via PBF, using a novel approach based on the use of miniaturized fatigue specimens. The designed miniature specimen was used to determine several fatigue characteristics, like the effect of the build orientation in fatigue performance [14], crack propagation [15] and the notch sensitivity of the material [16].

The main objective of the present work is to provide a mechanical characterization, focused on the high-cycle fatigue of Inconel 625 produced via laser DED. Additionally, the fatigue results obtained using cost-effective miniaturized specimens will be compared with standardized ones. The scarce information available regarding fatigue performance of DED Inconel 625 contributes to the originality of the present work, which provides useful information about the material to the final users and to the engineers involved in the design. Further a cost-effective fatigue testing methodology is explored which can be envisaged as a new trend for AMed materials fatigue testing to be possibly incorporated in future testing standards, after validation.

## 2. Materials and Methods

The present section provides details about the material and experimental testing procedure developed. Details about the manufacturing process, metallographic analysis, hardness, quasi-static tensile and fatigue testing will be discussed. Furthermore, a numerical model for stress calculation for the proposed testing procedure and its validation will be presented.

### 2.1. Samples Manufacturing

The samples were produced in two stages. Firstly, they were manufactured using Directed Energy Deposition technology, and lastly, they were post-processed by machining, without any further thermal or other post-processing technique (as-built conditions kept). All samples were manufactured from Inconel 625 alloy. The baseplate material used was a DIN 40CrMnMo7 steel. Figure 1a,b present some specimen extraction orientations in the as-built material condition, were the first one refers to the miniature specimens still attached to the baseplate, and the second refers to one tensile specimen, after the separation from

the substrate. Hereafter, the specimen geometries will be discussed in details. Figure 1c presents an overview of the DED system where can be observed the robot (Kuka, Augsburg, Germany), the deposition head, the reference table and the protection cell. Each sample was machined from a single deposited block. This was done in order to assure that all specimens (from a given geometry) were subjected to nearly the same thermal history, including cooling rates, assuring a very similar microstructure and residual stresses for all specimens.

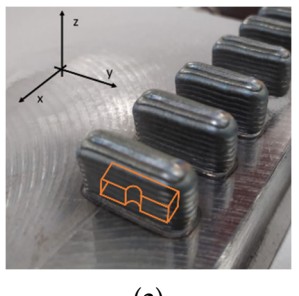
(**a**)

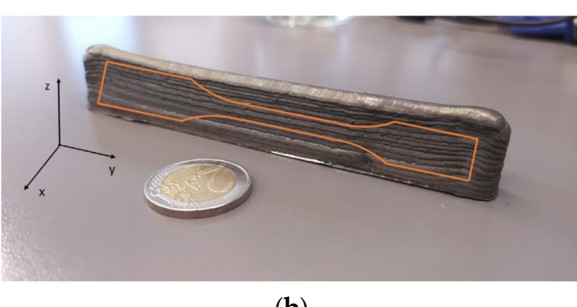
(**b**)

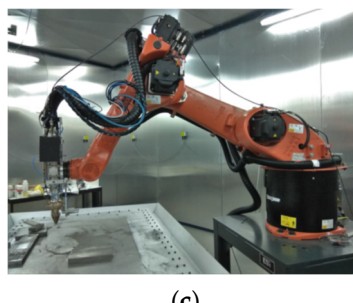
(**c**)

**Figure 1.** Samples manufacturing process: (**a**) Miniature specimen orientation; (**b**) Tensile specimen orientation [17]; (**c**) DED system [17].

All samples were produced using the same manufacturing parameters, summarized in Table 1, which were considered the optimized ones. The process used to find the optimized manufacturing parameters was an iterative try-and-error procedure, relying heavily on metallographic analysis. Preliminary samples manufactured using a set of trial parameters where inspected for defects (e.g., lack of fusion pores). Changes in the manufacturing parameters were performed if defects would be significant, and the procedure was then repeated. This procedure was kept until the final parts achieved a nearly defect-free condition.

**Table 1.** DED manufacturing parameters.

| Hatch Spacing (mm) | Layer Thickness (mm) | Laser Spot Diameter (mm) | Laser Power (W) | Powder Feed Rate (g/min) |
|---|---|---|---|---|
| 1.8 | 1.7 | 2.5 | 1800 | 12 |
| Scanning Speed (mm/s) | Shield Gas Flow Rate (L/min) | Carrier Gas Flow Rate (L/min) | Shield Gas | Carrier Gas |
| 6 | 26 | 2.5 | Argon | Argon |

### 2.2. Metallography

For each specimen geometry, a metallographic analysis was performed to the testing section. All samples were polished and observed on both etched and unetched condition. Additionally, samples from different planes were analysed. Unetched samples were used in order to inspect the porosity.

The etching process was performed by electrolysis using oxalic acid dihydrate, where the samples were placed on an electrolysis cell (Struers, Copenhagen, Denmark) and direct current was applied. This process was performed in order to reveal the deposited beads, layers and the microstructure.

### 2.3. Fatigue Tests

High-cycle fatigue tests were performed in two types of specimens. Figure 2a presents the miniature bending specimen, which were originally proposed by Nicoletto [18]. This geometry allows a significant reduction in the amount of material required for a fully established S-N curve, and consequently, reduces the costs involved. This is especially interesting for fatigue assessment of MAMed materials, once usually at least a dozen of specimens are required, and the costs per samples tend to be high. Also, such miniature

specimens could easily be extracted in any construction direction allowing new possibilities for material characterization. It is important to emphasize that in spite of the presence of a notch on the samples, the tests were performed in such a way to generate tensile stresses on opposite side and consequently the crack initiation will occur at that location. The notch presence aims only to assure that the central cross section will be the critical one, presenting the highest stresses.

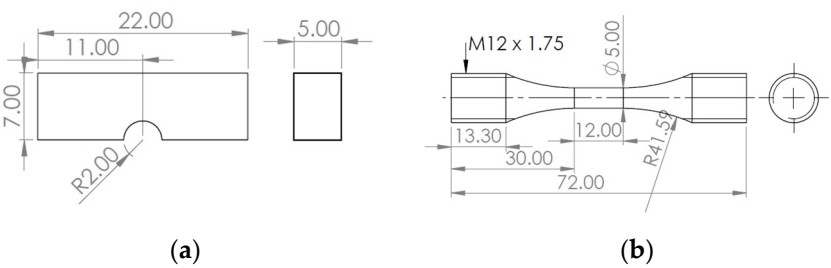

(a)                                                  (b)

**Figure 2.** Fatigue specimens geometry and dimensions in mm: (**a**) Miniature bending [17]; (**b**) Standard axisymmetric axial, adapted from [19].

Figure 2b presents the dimensions of a standard axial sample, which was designed in accordance with ASTM E466 [19], a standard for fatigue tests under force control. These tests will be used as reference values. Both miniature and standard specimens were tested for a load/stress ratio (R) of 0.1. The build orientation for both samples can be seen in Figure 3, where Z corresponds to the build orientation. Due to the process of baseplate separation and machining process, the minimum distance from the samples to the baseplate reference is 4 mm.

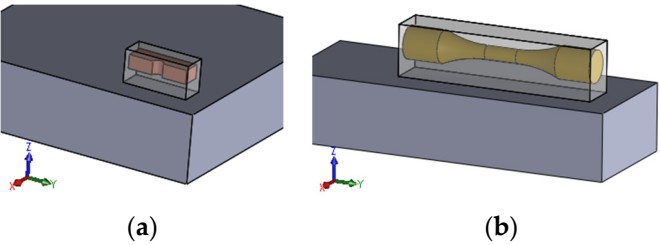

(a)                                                  (b)

**Figure 3.** Build orientation: (**a**) Miniature bending; (**b**) Standard axial (z corresponds to the build orientation).

The experimental setup for miniature bending and standard axial tests can be seen in Figure 4a,b, respectively. The tests of the standard specimens were performed under force/stress control on a servo-hydraulic machine (MTS System, Eden Prairie, MN, USA). On the other hand, the experimental apparatus for the Nicoletto type specimen (miniature bending) operates under displacement control. For these tests, a 4-point bending system was designed and developed in-house, and the specimen was hold in place by a gripping system. Section 2.5 will present additional details about the fixturing apparatus for the miniature specimens. In addition, it will provide details on the calibration of a numerical model, which encompasses the plasticity effect during testing, allowing the accurate prediction of the applied stresses.

During the bending fatigue tests, the reaction forces were measured by two load cells, and the maximum and minimum average values were used to calculate the stress on the specimen. The nominal stress can be calculated, assuming linear elastic behaviour, as:

$$\sigma_{nom} = \frac{M\,c}{I} \tag{1}$$

where $M$ is the bending moment, $c$ is the distance from the neutral axis (for the present case, half of the height of the central cross section) and $I$ is the second order moment of inertia for the central section (on the present case, a square section). The load ratio for

miniature bending tests scattered between 0 and 0.1. This little variation happened due to the displacement control, where (for a fix displacement range), sometimes the minimum force could reach values near 0 during testing. For the standard axial tests, the stress is simply calculated by:

$$\sigma_{axial} = \frac{F}{A} \tag{2}$$

where $F$ is the force, and $A$ is the cross-section area at the specimen centre.

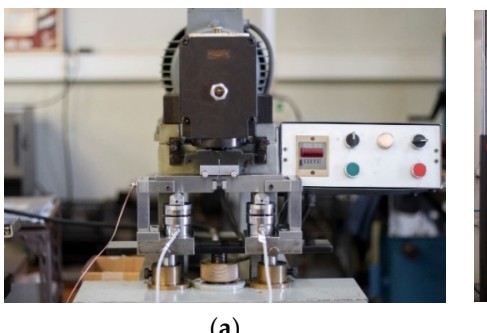
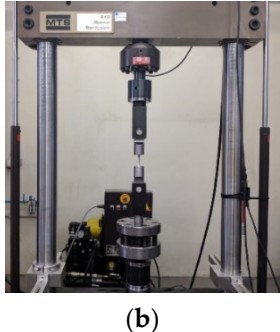

(a)　　　　　　　　　　　　　　　　　　　　(b)

**Figure 4.** Fatigue testing setup: (**a**) Miniature bending apparatus (pure bending); (**b**) Standard axial setup [17].

A comparison between the force (standard samples) and displacement control (miniature ones) tests will be performed. Even though the load mechanism of tests differ, they can be compared under certain circumstances. Prior to crack propagation, under the nucleation phase, the cross section of the miniature specimen is constant, resulting in a constant stiffness of the system. Therefore, for a displacement control, the load forces will have constant amplitude during this stage, allowing a direct comparison with the load controlled fatigue test. However, it is important to emphasize that this comparison is valid only at the nucleation stage.

Regarding the number of cycles to failure, for the miniature specimen tests, the number of cycles to crack initiation was determined corresponding to a reduction of 10% for the maximum force measured. This happened because, when the crack propagates, the effective cross section of the samples was reduced. Therefore, the overall stiffness of the system decreased, and for a constant displacement range, this resulted in a decline of the reaction forces. For the axial tests, the number of cycles to failure was established as the number of cycles to completely separate the specimen into two halves. In the case of force control testing, the number of cycles between macroscopic crack initiation and final fracture would be very similar, due to the fatigue crack propagation acceleration.

In addition, a comparison with literature will be performed. S-N curves from different manufacturing process will be presented. For each series, the S-N behaviour will be estimated by the following equation:

$$C = \Delta\sigma^m N \tag{3}$$

where $\Delta\sigma$ is the stress range, $N$ is the number of cycle to failure, $C$ is a constant (different for each material, manufacturing process and testing condition) and $m$ is the inverse slope coefficient. Since the literature presents values from several mean stresses, all S-N curves with load ratios other than R = −1 (fully reversed loading) will be converted to this ratio. The equivalent stress amplitude was calculated using the Smith-Watson-Topper (SWT) [20] mean stress correction:

$$\sigma_{ar} = \sigma_{max}\sqrt{\frac{1-R}{2}} \tag{4}$$

where $\sigma_{max}$ is the maximum stress.

In order to provide further information, two common fatigue design coefficients, the inverse slope coefficient, *m*, and the characteristic stress range at 2 million of cycles to failure, $\Delta\sigma_c$, can be established:

$$\Delta\sigma^m N = \Delta\sigma_c^m \times 2 \times 10^6 = C \tag{5}$$

The testing frequencies were 10 Hz for the miniature plane bending specimens and in the range of 6–8 Hz for the axial fatigue specimens. Load cells of 100 kN (MTS) were used in the servo hydraulic machine, while for the bending machine two equal load cells of (10 kN) were applied (U3 HBM).

The total volume of each miniature specimen corresponds to approximately 9% of the volume of an axial tensile standard specimen. Nevertheless, the miniature specimen cross area at the testing section (25 mm$^2$) is about 1.3 times higher than the axial tensile specimen testing section area (20 mm$^2$), which partially balances any size effect of the miniature specimen due to the non-uniform (bending) stress field.

### 2.4. Tensile and Hardness Tests

In order to perform a comprehensive mechanical characterization of the additive manufacturing material, tensile and hardness tests were performed. Regarding the first one, two specimens were tested, and they were designed according to the standard ASTM E8-21 [21]. Figure 5a presents the dimensions for the tensile specimens, while the build orientation is presented in Figure 5b. During tests, the applied force was measured by a load cell (HBM, Darmstadt, Germany), and the strains were measured by two distinct methods, Digital Image Correlation (DIC) and by an electrical strain gauge (Vishay Precision Group, Malvern, PA, USA). Due to the strain gauge setup limitation, only measurements up to 0.5% were performed with it, data above this limit were measured only via DIC.

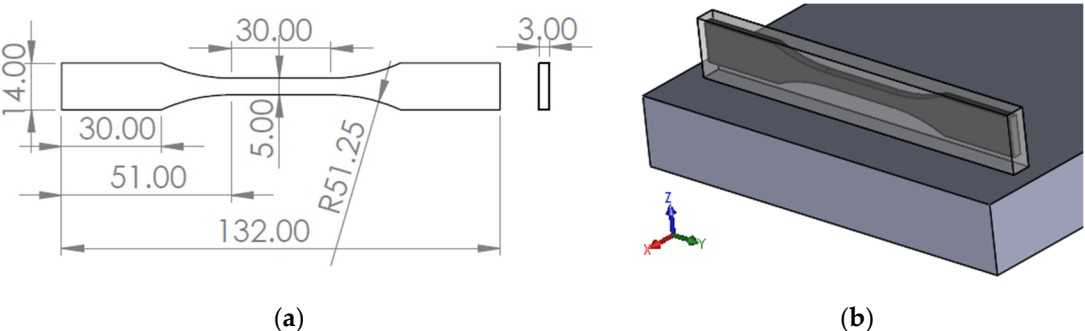

(**a**)                    (**b**)

**Figure 5.** Tensile specimen: (**a**) Dimensions, in mm; (**b**) Build orientation (z).

Vickers hardness measurements were performed on a specimen attached to the baseplate and, consequently, both the MAMed material and base plate were tested. Figure 6 pictures the sample were the indentations were performed. Measurements were performed 2 mm apart from each other, following the build direction (z). The hardness was measured on an as-built geometry, cut and polished. The geometry was the same as the ones used for obtaining the miniature bending specimens. For the present work, in spite of the focus being the Inconel 625 hardness, the measurements on the baseplate will also be discussed. This can be relevant for other investigations, since the deposition affects the baseplate microstructure and mechanical properties.

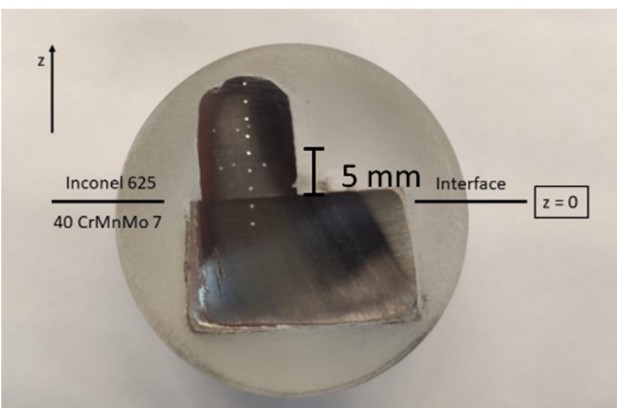

**Figure 6.** Hardness measurements on a specimen [17].

### 2.5. Numerical Model of the Miniature Specimen Fatigue Test Setup and Calibration

In order to use an existing four-point fatigue testing machine, a special gripping device was designed (manufactured in-house). Its main functions are to transfer the load to the miniature specimen and hold it in place during testing. It is important to understand how applied controlled displacements will be converted into stresses at the specimen gauge section. For that purpose, an elastoplastic numerical model of the testing setup was developed and calibrated using experimental stress analysis, since Equation (1) is only valid for elastic conditions and due to the high strain hardening of the material (high yield to ultimate tensile strength differential), finite fatigue lives would likely involve some plasticity. Therefore, the present section describes the numerical model and respective experimental procedure used for its calibration. The calculation of the stresses to be used for S-N curves determination from miniature bending samples will be detailed. Figure 7a presents the main dimensions of the assembled system. Figure 7b shows a strain gauge attached to the tensile face of the specimen which will be used to measure the tensile strains along the specimen's larger dimension (longitudinal direction).

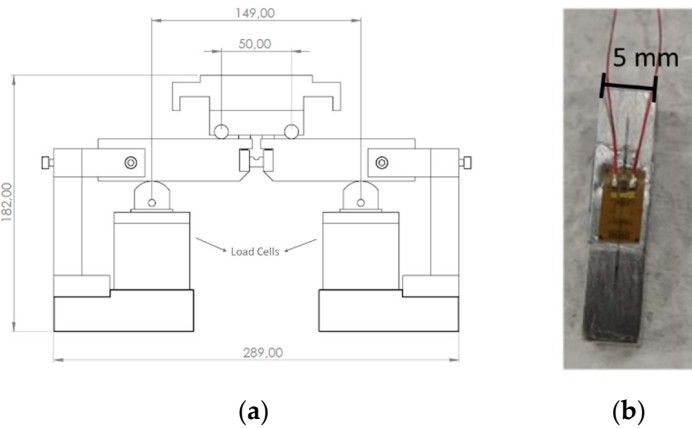

(**a**)          (**b**)

**Figure 7.** Calibration setup: (**a**) Four-point bending system, dimensions in mm [17]; (**b**) Strain gauge on specimen (reference MMF307385).

The experimental procedure consisted in incrementally increasing the external load, and for each loading step, the strain and reaction forces being measured. The strains were measured and the reaction force (for each step) was an average of both load cells measurements.

The main parts of the numerical model can be seen in Figure 8. ¼ of the setup was modelled benefiting of the existing two planes of symmetry, which reduced the modelling computational requirements and facilitated the boundary conditions. In order to properly represent the test, an external displacement was applied at the actuator pin. The reaction force in the y-direction was measured at the support. Additionally, in order to compare

with experimental data, a rectangular area with the same dimensions of the strain gauge grid was created in the model, and the average strain in the x-direction was used and compared with experimental measurements. The strain gauge grid dimensions were 1.52 mm × 1.27 mm, along x and z-directions, respectively.

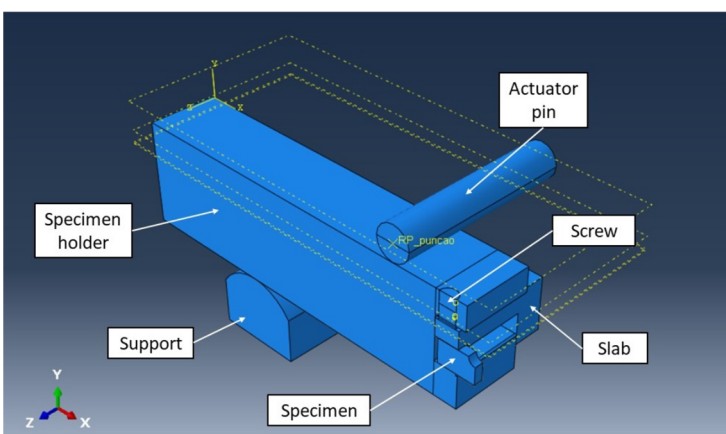

**Figure 8.** Numerical model main components [17].

In order to hold the specimen in place, a pre-load was applied to the bolt, similar to the experimental apparatus. Due to this load, the bolt applies a pressure on the slab (Figure 8), which holds the specimen. All adjacent components interactions were modelled using a contact approach, which allowed sliding between contact pairs, and a friction coefficient was applied.

Regarding the numerical model, linear 8-noded brick elements, fully integrated, were used. For material properties, all parts properties but the specimen were assumed linear elastic. For the specimen, a plasticity model was used, and the properties were based on the experimental tensile properties, explained in Section 2.4, with results shown in Section 3.2.

In order to calibrate the numerical model, small changes in some parameters were performed. These variations were performed within a range that respected the experimental uncertainties of some parameters (changes smaller than 5% were performed). The parameters adjusted were Young's modulus of specimen and gripping system, bolt pre-load and friction coefficients. Once a good agreement between numerical and experimental data (load vs. specimen strain monotonic test data) was achieved, the model was considered validated.

## 3. Results and Discussion

The present section describes the results regarding the microstructure and porosity analyses, followed by the mechanical characterization of the material, namely the tensile and hardness measurements. After, the results from the numerical model and its experimental validation will be provided. Lastly, the results from the fatigue tests on conventional and miniature specimens will be presented together with fracture surface analysis and a comparison with literature will be discussed.

### 3.1. Microstructural Investigation

Analysis of samples in both polished and etched conditions were performed. Figure 9a,b show a non-etched view of the samples. It can be seen that the samples have a significant amount of porosity near the baseplate (highlighted by red circles). However, for the present case, this is not a concern because this region will not be part of the final specimen, since 2 to 3 mm of material thickness was removed due to baseplate separation process. At the centre of the specimen (testing section), it can be seen that the sample is nearly 100% dense, and very small porosities are seen, mainly resulting from entrapped gases. The measured porosity from Figure 9b was 0.075%, resulting a very high relative density of 99.925%.

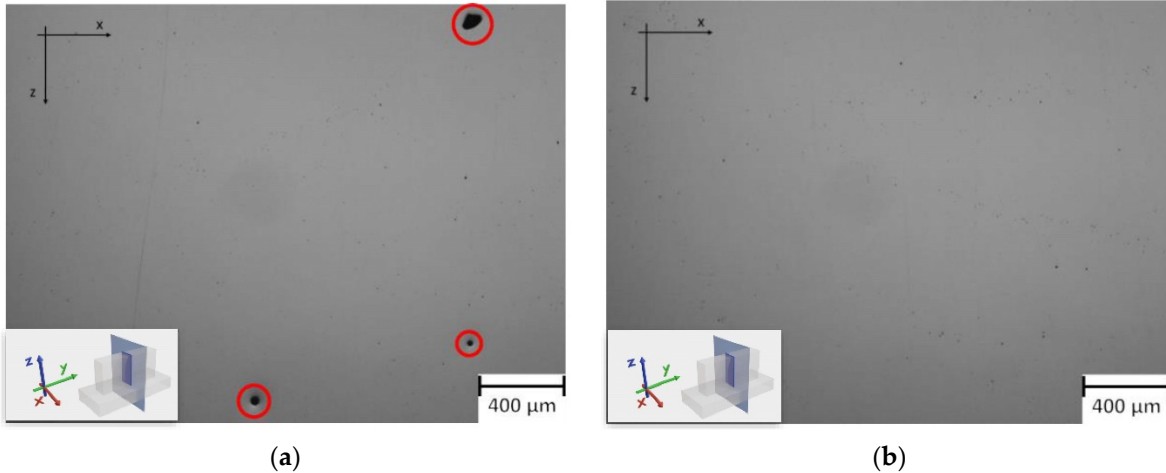

(**a**)　　　　　　　　　　　　　　　　　　　　　(**b**)

**Figure 9.** XZ surface porosity: (**a**) Near baseplate interface (pore zones in red) [17]; (**b**) Intermediate height (mechanical testing region) [17].

Figure 10a presents an overview of an etched sample (as-built cross section, which will lead to a standard axial fatigue specimen). Each layer can be identified, as well as the beads. Figure 10b shows a zoomed-in view, in which the hatch spacing and layer thickness were measured. The results were in good agreement with the analytical data, input for the G-code, i.e., the Computer Numerical Control program, which includes information regarding laser trajectory and other relevant process parameters. The measured hatch spacing was 1.78 mm, very similar to the 1.80 mm from the reference value (Table 1). The measured value for layer thickness, 1.75 mm, is also very close to the inputted value, 1.7 mm. Regarding the metallurgical properties, it can be seen a good bonding between layers and beads, with no lack of fusion defects in the interfaces. In addition, dendrites can be identified, and often some of them extends across multiple deposited tracks.

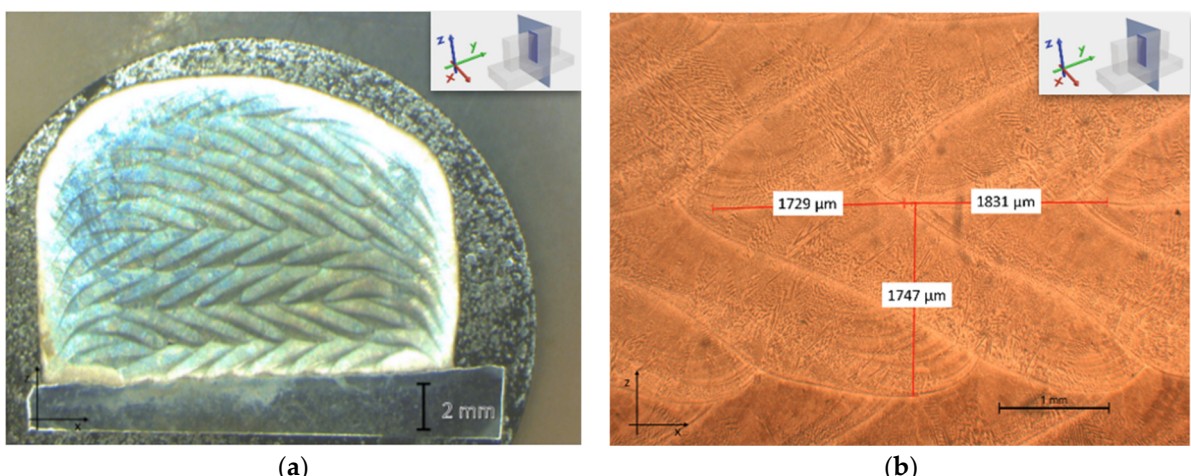

(**a**)　　　　　　　　　　　　　　　　　　　　　(**b**)

**Figure 10.** Etched XZ surface cut: (**a**) Sample overview [17]; (**b**) Measured hatch spacing and layer thickness [17].

### 3.2. Tensile Testing and Hardness Results

In order to properly characterize the mechanical properties, tensile and hardness tests were performed. The monotonic Stress-Strain curves for specimens 1 and 2 can be seen in Figure 11. The results for both tests were very similar, with the Specimen 1 displaying a little more elongation.

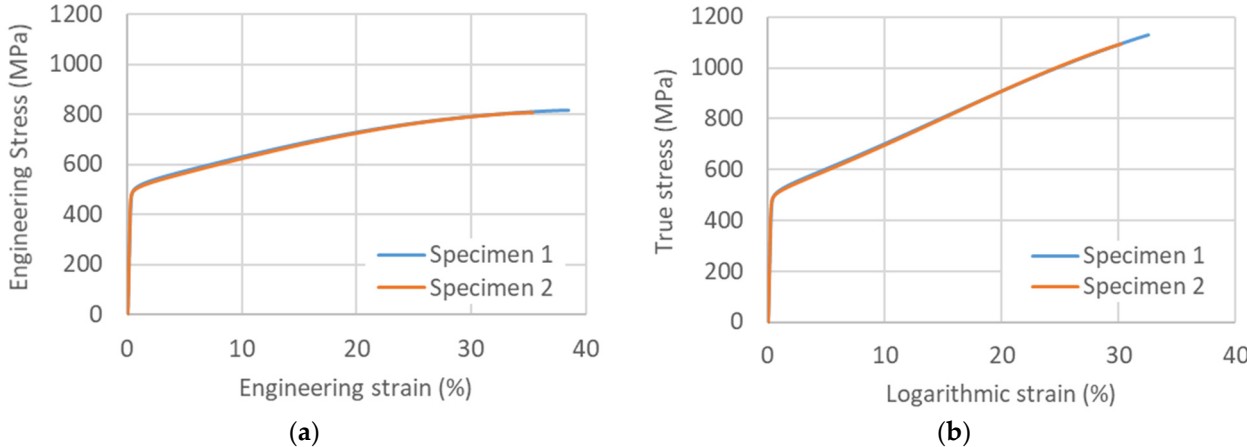

**Figure 11.** Stress-strain curves: (**a**) Engineering; (**b**) True.

The tensile properties are presented in Table 2, and they are an average of both tests. Comparing the yield stress and the ultimate tensile strength (UTS) it is clear a very pronounced strain hardening, which could be more than 100% when considering the true strength values. The tensile elongation was higher than 30%, for both samples.

**Table 2.** Tensile properties measure for the MAMed Inconel 625.

| Young Modulus (GPa) | Yield Stress, 0.2% (MPa) | Engineering Ultimate Tensile Strength (MPa) | True Ultimate Tensile Strength (MPa) |
|---|---|---|---|
| 190 | 496 | 810 | 1100 |

The Vickers hardness were also measured on a sample. Measurements were performed on several z-coordinates, as shown in Figure 6. The results are shown in Figure 12. Negative z-coordinate values correspond to measurements at the baseplate. A pattern can be noticed for Inconel 625, where the hardness decreases with the increase of the distance to the base plate (z). This can be explained based on the cooling rates and resulting microstructure. The first layers built presented higher cooling rates than the last ones, once they are closer to the baseplate and heat can be easily dissipated. Higher cooling rates tend to result in finer grain structures, and higher hardness. Therefore, it was expected that the layers which experienced faster cooling rates (first layers) were the ones with the higher hardness.

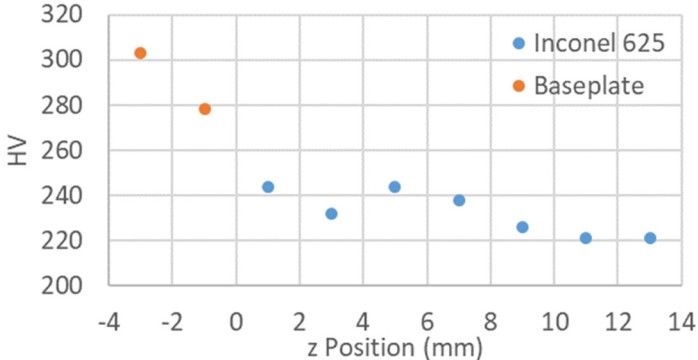

**Figure 12.** Vickers hardness.

The hardness and tensile properties were compared with literature values, for wrought, DED and SLM specimens [11], shown in Table 3. The properties of the present work represent the averages of all tests performed, and these values will used for comparison (deviation). Positive values of deviation imply that the studied material showed superior

properties than the literature values. The investigated material presented a yield stress similar to the wrought material, but lower than the other AMed materials. In addition, the studied material demonstrated the lowest UTS (engineering stress) among the four considered cases. On the other hand, the hardness was the lowest and the elongation was lower than the wrought material but well above the other AMed materials. It is interesting to note that in general, the major differences were found between the current material and the literature DED based material, which is a clear indication of the key influence of the processing parameters on material properties.

**Table 3.** Comparison of tensile and hardness properties.

| Material | Young Modulus (GPa) | Deviation (%) | Yield Stress, 0.2% (MPa) | Deviation (%) | Ultimate Tensile Strength (MPa) | Deviation (%) |
|---|---|---|---|---|---|---|
| Present Work | 190 | - | 496 | - | 810 | - |
| Wrought [11] | 184 | 3 | 482 | 3 | 955 | −18 |
| SLM [11] | 245 | −29 | 652 | −31 | 925 | −14 |
| DED [11] | 223 | −17 | 723 | −46 | 1073 | −32 |
| **Material** | **Elongation (%)** | **Deviation (%)** | **HV** | **Deviation (%)** | **-** | **-** |
| Present Work | 37 | - | 232 | - | - | - |
| Wrought [11] | 41 | −11 | 260 | −12 | - | - |
| SLM [11] | 32 | 14 | 313 | −35 | - | - |
| DED [11] | 26 | 30 | 315 | −36 | - | - |

*3.3. Numerical Model Calibration*

As detailed in Section 2.5, the numerical model was calibrated in order to accurately represent the test setup, and the procedure was supported by experimental data, namely the strain at specimen gauge section and reaction forces.

Figure 13a presents a comparison between the final numerical model results and the experimental data. A good agreement was found, especially for lower reaction forces. The largest discrepancy between numerical and experimental data, in terms of local elastoplastic strains, was 5.4% for an applied reaction force of 429 N (biggest measured experimental reaction). It can be noticed that for reaction forces around 250 N (resulting on a nominal stress around 500 MPa, according to Equation (1)), the graphic is no longer linear. For reaction forces of this magnitude, the testing section experiences plasticity, once the stresses on this region are above the yield stress, resulting in the non-linearity between reaction forces and strain.

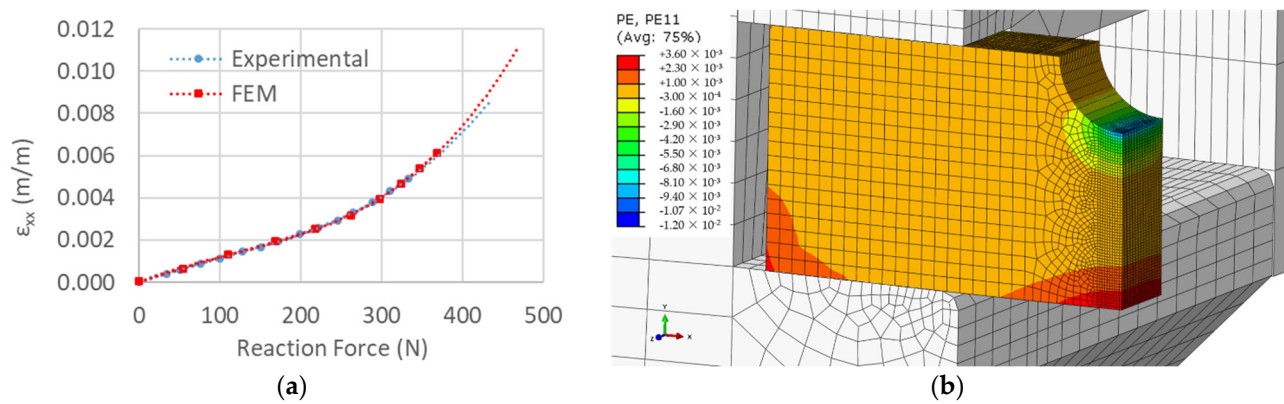

**Figure 13.** Numerical results: (**a**) Numerical and experimental comparison; (**b**) Normal plastic strain in x-direction, external force of 470 N.

The plastic normal strain in x-direction after a full load (maximum external displacement applied of 0.75 mm) can be seen in Figure 13b. The largest plastic deformation can be

seen at the symmetry plane on x-direction. As expected, due to the stress concentration, the notch presented the largest plastic strain (due to the presence of the largest stresses). However, since these stresses are always compressive, they do not lead to fatigue cracking during tests.

The final parameters used in the numerical model calibration can be seen in Table 4. The gripping system refers to all parts excluding the specimen. This encompass the support, actuator pin, bolt, support and slab (as shown in Figure 8). The pre-load value used was approximately the one resulted from the binary of 10 N·m applied to the bolt, prior to fatigue tests.

**Table 4.** Parameters used on the calibrated numerical model.

| Young Modulus of Gripping System (GPa) | Young Modulus of Specimen (GPa) | Bolt Pre-Load (kN) | Friction Coefficient | Bolt Nominal Diameter (mm) |
|---|---|---|---|---|
| 190 | 196 | 10 | 0.15 | 5 |

The plastic strain and stresses used for the numerical model are presented on Table 5 and were obtained from the monotonic stress-strain curve. A Mises yield criterion was used with multilinear kinematic hardening. When the material is loaded above its yield stress during fatigue tests, it will be subjected to plastic deformation. The maximum peak strain in all test programme would be around 0.9%, according to Figure 13a results. This will achieved in the first fatigue cycle, during the loading stage. After, the specimen will be unloaded up to a load value of 10% of the maximum value (applied stress R-ratio of approximately 0.1). The plasticized material will experience elastic unloading and due to the applied load ratio value and limited maximum plastic strain level and consequent hardening, an elastic shakedown condition will take place even if the local stress ratio would be slightly negative.

**Table 5.** Plastic strain and stress used on the numerical model.

| Plastic Strain (%) | True Stress (MPa) |
|---|---|
| 0.00 | 496 |
| 0.29 | 505 |
| 0.49 | 513 |
| 0.99 | 532 |
| 1.79 | 551 |
| 7.49 | 665 |
| 29.69 | 1100 |

### 3.4. High-Cycle Fatigue Behaviour

Regarding high cycle fatigue (HCF) tests, 26 specimens were tested, being 10 of them standard axial samples, and 16 miniature bending ones. The S-N curve can be seen in Figure 14. The specimens that did not experience failure (run-outs) are identified using arrows. It is clear that the nominal stresses for miniature bending specimens, computed assuming unbounded elastic stresses (Equation (1)), overestimates the fatigue strength, sometimes presenting stress levels above the engineering UTS (pseudo elastic stresses). This is mainly due to the linear elastic hypothesis employed in these calculations, which do not allow a perfect match with the stress-controlled axial fatigue tests. Therefore, using an approach able to encompass plasticity is essential, and this can be done using the validated FEM model. Regarding this approach, the S-N results from the miniature specimens are much more consistent with the reference values (standard axial samples), confirming the new test procedure based on miniature specimens.

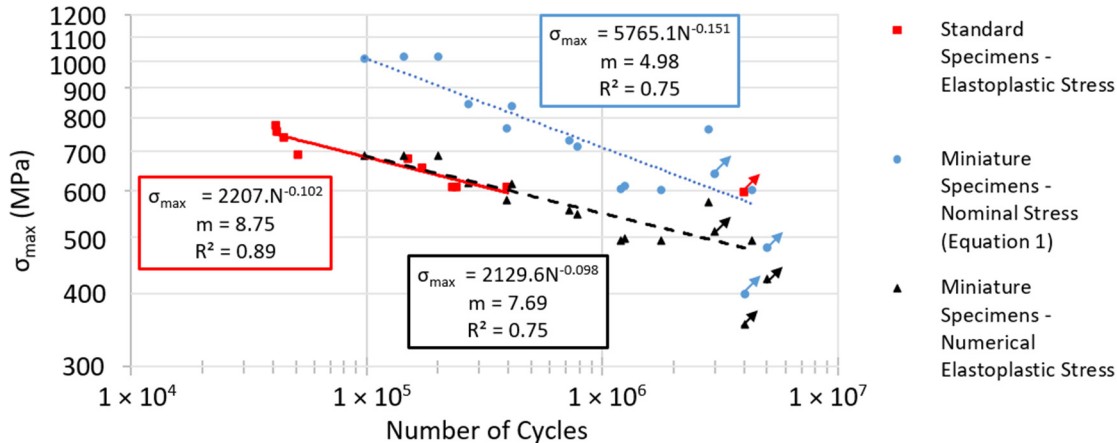

**Figure 14.** S-N curves, standard and miniature specimens, R = 0.1.

Due to the good agreement between the two tests data, both miniature (numerical elastoplastic stresses) and standard specimen results were combined, in order to facilitate the comparisons with literature. The resulting S-N curve can be seen in Figure 15, demonstrating a high determination coefficient. As referred in the Section 2.4, the miniature specimens show a cross section slightly higher than the axial tensile specimens, which seems to compensate any expected higher fatigue strength for the miniature specimens in the case of similar cross sections. Also, the plasticity would have a smoothing effect of the linear stress distribution of the miniature specimens reducing the possible size effects. Despite in this research size effects are not visible, they have been reported in the literature when axial and miniature specimens show similar cross sections and elastic loading is observed. In those cases, a multiplicative correction factor of 0.91 is applied to the miniature fatigue test data [16].

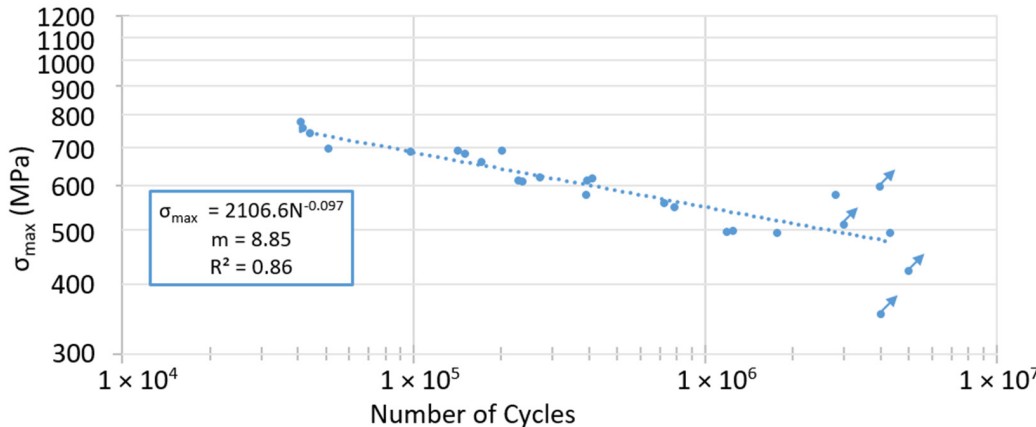

**Figure 15.** S-N curve, grouped standard and miniature specimens, R = 0.1.

The obtained results were also compared with literature. It is important to emphasize that the present material was not heat-treated or polished, which could result on even higher fatigue properties. Figure 16 presents a comparison of the values obtained in this work with literature.

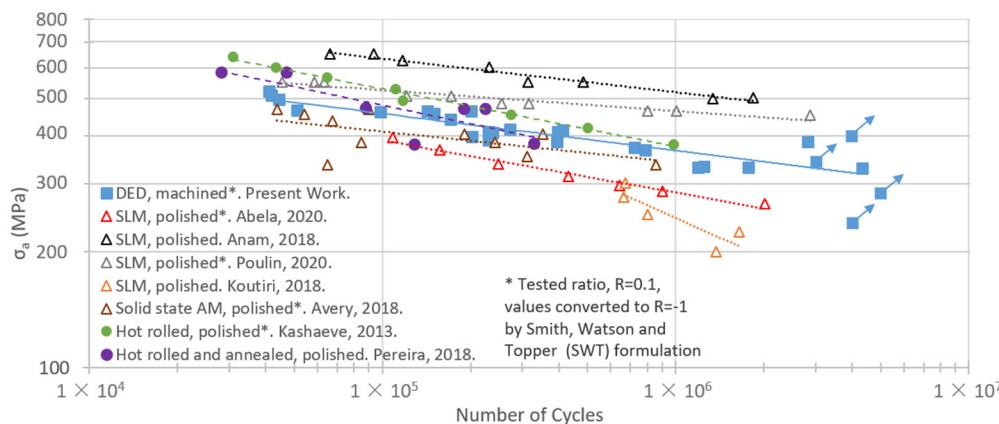

**Figure 16.** Literature comparison, S-N, R = −1, data from [22], [23], [24], [25], [26], [27] and [13], respectively (according to the sequence presented in the chart legend).

Since literature data was obtained for different stress R-ratios, the results were compared using the SWT correction according to Equation (4). As a direct comparison, the presented DED Inconel 625 presented fatigue properties better than 3 out of 5 ([22,25,26]) AMed materials (Selective Laser Melting and Solid State). The two AM materials with superior properties were obtained via SLM and were polished ([23,24]). When comparing with conventional materials, the DED Inconel 625 was similar to a hot rolled and annealed material [13] (presenting lower fatigue strength near the low-cycle fatigue regime, and better performance for an higher number of cycles), and lower fatigue strength when compared with the conventional material on the hot rolled and polished condition [27].

The inverse slope (*m*) and stress range at 2 million cycles ($\Delta\sigma_c$) can be seen in Table 6, as well as variations with respect to the present study. The data for materials marked by an asterisk were converted from R = 0.1 to R = −1. These fatigue parameters are dependent on several factors, like metallurgical (for example type of alloy, microstructure, grain size and defects), surface finishing, testing temperature and load ratio. Small values of the inverse slope indicate that the fatigue performance of the material reduces drastically when the number of cycles is increased. The inverse slope could reflect the impact of crack initiation and crack propagation phases. When crack propagation dominates, the inverse slope should present low values (between 3 and 4). When the fatigue crack initiation is the dominant damage regime (which is the present case), inverse slope, *m*, should be higher and sensitive to many parameters. The $\Delta\sigma_c$ is often used as project parameter, because most of components are subjected to number of cycles inferior of this limit. 4 results were poorer than the current study, one was very similar and two were better illustrating a very satisfactory performance of the produced material.

**Table 6.** Fatigue properties comparison.

| Material | *m* | $\Delta\sigma_c$ **(MPa)** | $\Delta\sigma_c$ **Deviation (%)** |
|---|---|---|---|
| DED, machined (present work) | 8.85 | 692 | - |
| SLM, polished * [22] | 7.23 | 516 | −25.4 |
| SLM, polished [23] | 11.09 | 970 | 40.2 |
| SLM, polished * [24] | 17.15 | 884 | 27.7 |
| SLM, polished [25] | 2.27 | 358 | −48.3 |
| Solid state AM, polished [26] | 4.65 | 450 | −35.0 |
| Hot rolled, polished * [27] | 6.60 | 672 | −2.9 |
| Hot rolled and annealed, polished [13] | 4.07 | 464 | −32.9 |

Figure 17a shows the fracture surface of a standard axial fatigue specimen, which was submitted to the highest tested stress of 777 MPa. The crack nucleation region can be seen at the top of the picture, and is displayed with more details in Figure 17b. This area was characterized by a very uneven fracture surface, where the crack propagation did not follow

the global direction of maximum stress. Instead, the microstructure was the dominant factor for fatigue, where favourable crystallographic planes led the crack propagation. In this region, it was not possible to identify fatigue striations. If the crack growth rate is so small, the striations are not visible (they are very close to each other). This could occur when the fatigue crack propagation regime is in the near threshold regimes. Also the striations observation depends on the complex crack-microstructures interactions. For the analysed samples, the crack initiation region (Figure 17b) presented a Crystallographic Growth Region (CGR), this area was characterized by a very uneven surface, where the crack propagation did not follow the global direction of maximum stress. Instead, the microstructure was the dominant factor for fatigue, where favourable crystallographic planes led the crack propagation. The red square on the miniature images (top of each figure, when present) indicates the position of the image on a global view of the sample.

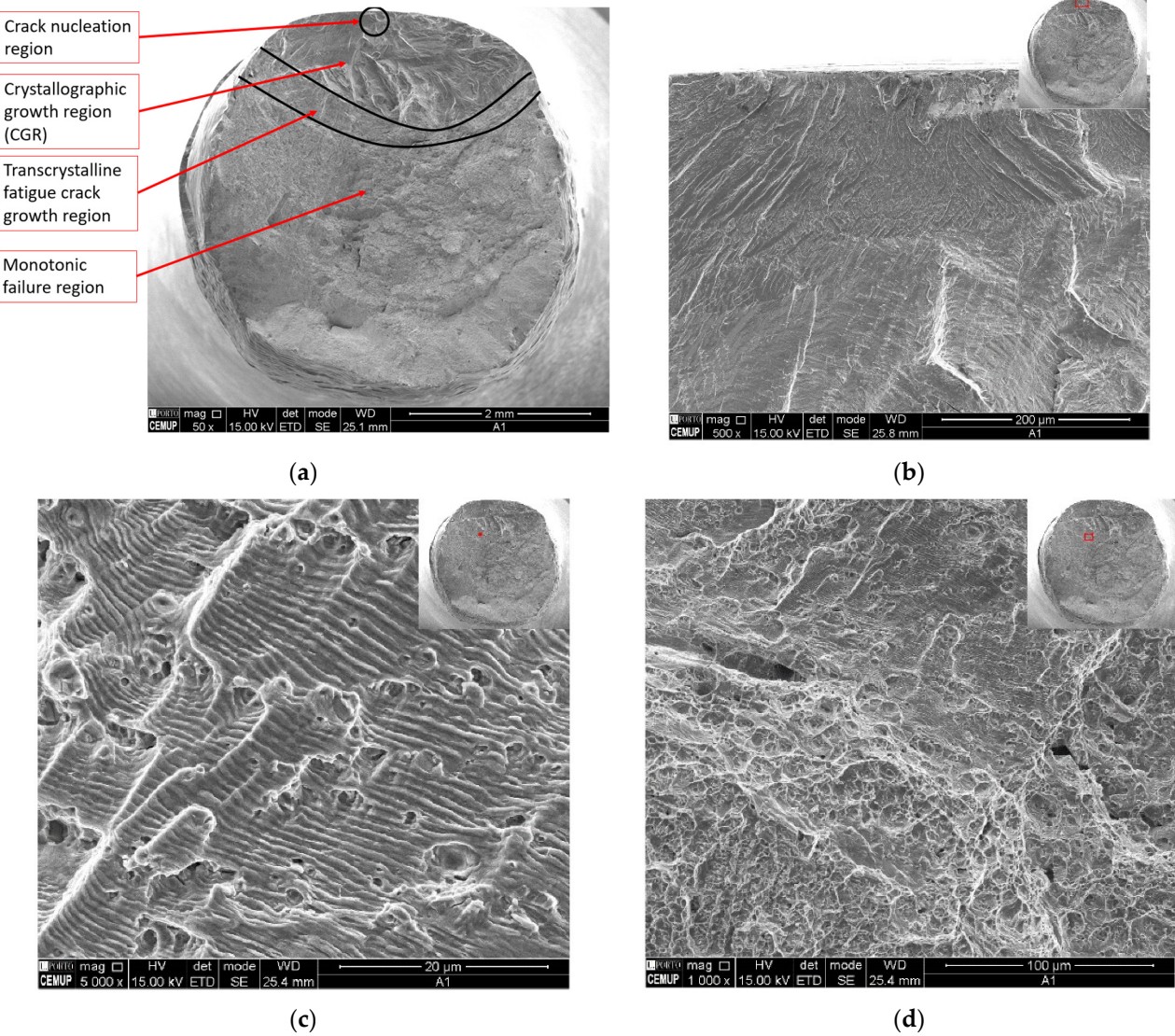

**Figure 17.** Fracture surfaces for standard axial fatigue specimen, sample A1, maximum stress = 777 MPa: (**a**) Fracture surface overview; (**b**) Crack nucleation region; (**c**) Transcrystalline fatigue crack growth region; (**d**) Interface between fatigue and monotonic failure.

The striations are easier detected when the crack propagates on a constant direction (orthogonal to the loading direction), as is less influenced by the material microstructure, as can be seen in Figure 17c. At this stage, a macroscopic crack is present, and the crack growth rates have a significant magnitude. The crack propagation followed a nearly constant

direction (small changes in direction occurred due to pores and other microstructure features). Figure 17d presents a transition between the fatigue (at the top of the figure), where the striations are still visible, and monotonic ductile failure (bottom portion of the figure). Due to the high maximum stress during this test (777 MPa), the monotonic failure corresponds to a significant amount of the total cross section.

Figure 18 presents fractures surface of Sample A6, also an standard axial fatigue specimen. The maximum stress on this sample was lower than the previous analysed sample (A1), 658 MPa. Figure 18a presents the nucleation site. Again, a CGR can be identified at the crack initiation region. In this sample, it was also possible to identify striations near the centre of the specimen, seen in Figure 18b. When comparing with conventional Inconel 625 [13], the fracture surfaces differ a lot in the nucleation zone. This happens mainly because, as stated before, at this stage, the microstructure has a leading role on the crack propagation. However, the fracture surfaces are much more similar for higher crack growth rates, and in both AMed and conventional material, the fatigue striations are visible at this stage.

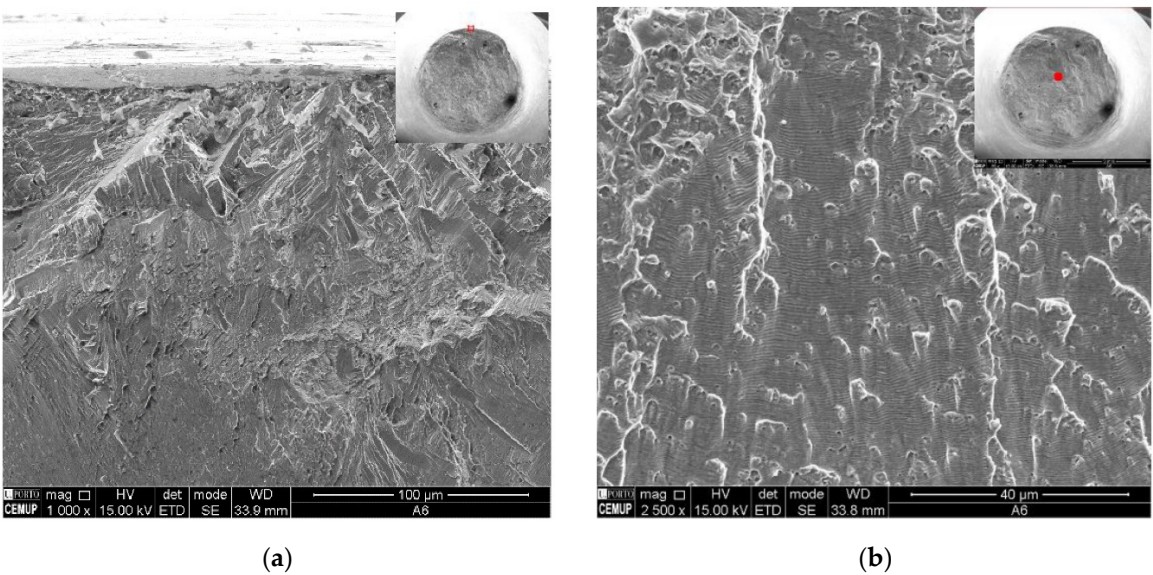

(**a**)                                                (**b**)

**Figure 18.** Fracture surfaces for standard axial fatigue sample, sample A6, maximum stress = 685 MPa: (**a**) Crack nucleation region; (**b**) Transcrystalline fatigue crack growth region.

The fractured surfaces for miniature specimen were also analysed. Figure 19 presents the fracture surface of Sample N7 (miniature specimen), where the maximum stress was 615 MPa, according to the calibrated FEM model (or a nominal maximum stress of 845 MPa). The miniature tests do not have a constant stress at the cross section of the specimen (as happened on the standard axial ones), they have a tensile stress at the top surface, indicated in Figure 19a, and compressive stress at the notch (lower region of the figure). Once more, the striations marks are seen nearly the centre of the specimen, Figure 19b, and are not noticeable at the nucleation regions. The analysis of the fracture surfaces do not show evidence of the presence of defects at the origin of the fatigue crack initiation, which is very common on the SLM processes.

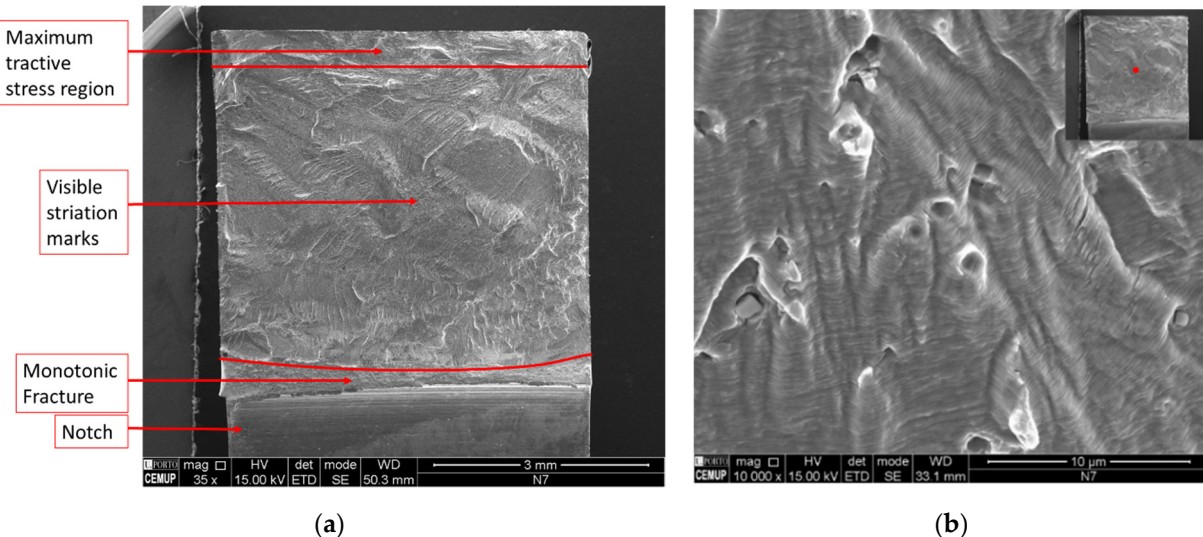

**Figure 19.** Fracture surfaces for miniature sample, N7, maximum FEM stress = 615 MPa: (**a**) Overview; (**b**) Striation marks view.

## 4. Conclusions

The miniature specimen proposed for fatigue characterisation provided a sound agreement with standard uniaxial tests. Therefore, it can be stated that, for the present analysis, the miniature specimens are able to successfully characterize the high-cycle fatigue behaviour. These results are achieved with a very small fraction of the required material and costs, if compared with standard tests. However, it is important to emphasize that a validated numerical model is required, mainly in cases where elastoplastic behaviour will occur in order to lead to a finite life. This is the case of the Inconel 625, which shows a very high monotonic strain hardening behaviour. Fatigue lives below 1E6 cycles (R = 0.1) would require maximum stresses surpassing the yield strength of the material in the first load cycle.

The optimized DED manufacturing parameters used for specimen production showed to be efficient, and able to produce parts with a limited content of pores. Regarding the material properties, the Inconel 625 manufactured via DED presented good fatigue performance, being able to bear hundreds of thousands of cycles in stress levels above its yield stress (for a load ratio of 0.1). When comparing the HCF results with literature, the Inconel 625 from the present work has been found to have properties similar to the conventional material, in the annealed condition. In addition, it has showed superior properties than some SLM specimens.

**Author Contributions:** Fatigue testing: F.K.F., D.M., M.F.; numerical model: F.K.F., D.M., J.G.; writing; F.K.F., D.M., J.G., F.B.; project administration: A.d.J.; experimental data acquisition: D.M., M.F.; conceptualization: F.K.F., A.d.J., F.B.; methodology: A.d.J., F.B. All authors have read and agreed to the published version of the manuscript.

**Funding:** This research was funded by FEDER and National Funds (FCT), Reference PTDC/EME-EME/31307/2017, through the Add.Strength project entitled "Enhanced Mechanical Properties in Additive Manufactured Components". The authors acknowledge the funding of a PhD scholarship with the reference UI/BD/150684/2021, funded by FCT, and the project with reference LAETA–UIDB/50022/2020 and UIDP/50022/2020 by FCT as well. In addition, this research was also funded by FEDER and National Funds (FCT), Reference PTDC/EME-EME/7678/2020, through the GCYCLEFAT project entitled "Giga-Cycle Fatigue Behaviour of Engineering Metallic Alloy".

**Conflicts of Interest:** The authors declare no conflict of interest.

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
