# Peer review of "Fatigue Assessment of Inconel 625 Produced by Directed Energy Deposition from Miniaturized Specimens"

_metals, doi:10.3390/met12010156_

Round 1

Reviewer 1 Report

The paper presents an S/N (stress-life) fatigue assessment of Additive Manufactured via DED Inconel 625 samples. The work is focused on demonstrating that a four-bend fatigue testing method using miniaturized specimens is capable of reproduce the S/N trend obtained by conventional uniaxial fatigue tests. The developed work is in general interesting. However, while the experimental part is satisfactorily described, there are some gaps in the theoretical and numerical development that must be addressed before the paper is considered for publication: 1) A deepening of conceptual discussion should be included in paper regarding the differences between the two fatigue testing methods, beyond the specimen size. First, it is known that, even under load control, axial and bending fatigue tests result in distinct S/N curves, the baseline fatigue strength in bending being higher than in axial loading (due to the non-uniform stress distribution in bending). Moreover, in addition to having different types of loading, the tests were performed under distinct control modes and cycle asymmetry (R = 0.1). Under these conditions, creep-fatigue interaction may occur in load-controlled tests and stress relaxation in displacement-controlled tests. How valid is (conceptually) the comparison between these tests? Such a perfect match between standard and miniature specimens S/N curves (Figure 14) is to be expected, based only in the correction for elastoplasticity? 2) The authors developed a numerical analysis in order to correct the bending fatigue results for plasticity. In my opinion, the strain-hardening equation (plasticity model) used in non-linear FEM analysis should be explicitly presented in paper as well as any other assumption necessary to calibrate the numerical model, that is, superpose the FEM results to the experimental strain measurements. 3) The authors claim that the plasticity model used was based on the experimental tensile properties. Since there may be differences between the monotonic and cyclic stress-strain curves for a material, some additional discussion on the cyclic behavior of DED-processed Inconel 625 should be included in paper. This reviewer also points out some minor corrections to be done:   

- line 132: “… seen in Figure…”

- line 150: “… half of the height…”

- line 243: “… achieved, …”

- line 267: Please explain what is the G-code.

- Figure 11: Please consider to write “Logarithmic strain” instead “True strain”.

- Table 2 and Table 3: please use “Yield Stress” instead of “Yield Strenght”.

- line 302: “… and the yield and UTS were the under the…”? I did not understand this statement.

- line 365: please write “axial fatigue specimen” instead of “tensile specimen”.

I conclude this review by stating that it is a quite interesting paper that should be published after major revision.

Author Response

The paper presents an S/N (stress-life) fatigue assessment of Additive Manufactured via DED Inconel 625 samples. The work is focused on demonstrating that a four-bend fatigue testing method using miniaturized specimens is capable of reproduce the S/N trend obtained by conventional uniaxial fatigue tests. The developed work is in general interesting. However, while the experimental part is satisfactorily described, there are some gaps in the theoretical and numerical development that must be addressed before the paper is considered for publication:

1) A deepening of conceptual discussion should be included in paper regarding the differences between the two fatigue testing methods, beyond the specimen size. First, it is known that, even under load control, axial and bending fatigue tests result in distinct S/N curves, the baseline fatigue strength in bending being higher than in axial loading (due to the non-uniform stress distribution in bending). Moreover, in addition to having different types of loading, the tests were performed under distinct control modes and cycle asymmetry (R = 0.1). Under these conditions, creep-fatigue interaction may occur in load-controlled tests and stress relaxation in displacement-controlled tests. How valid is (conceptually) the comparison between these tests?

A: The comments are very pertinent and helped to increase the discussion in the paper. A discussion about the comparison between the displacement and force-controlled tests was improved in the manuscript. In addition, the transferability between tests was discussed – the so called size effect was discussed. Finally, some considerations about the cyclic plasticity were added. In short, the displacement controlled tests were carefully analysed and the possible relaxation in the displacement controlled tests due to damage was discarded; also any local relaxation due to the plastic action was also discarded by simulation due to elastic shakedown after the first cycle. For the axial tensile specimens, the referred possible creep-fatigue would be only viable if there would be some degree of reversibility in the loading which is not possible with the R=0.1. Size effects are discussed (compared specimens cross sections) and some literature results referred.

Such a perfect match between standard and miniature specimens S/N curves (Figure 14) is to be expected, based only in the correction for elastoplasticity?

A: That is an interesting question and it seems that there are some contra balancing aspects regarding the size effects. Higher cross section in the miniature specimen with some plasticity would be the reason. A quick summary (in chronological order) of the activities leading to the interesting result is listed as follows:

  • Tensile tests were performed
  • Fatigue tests (S-N curves) were performed for miniature specimens.
  • For the miniature fatigue tests, the nominal stresses (Figure 14, blue series) were mainly above the yield stress. Therefore, it was concluded that using a linear-elastic approach (nominal stresses) does not provide a proper estimative of stresses (some nominal stresses were even above the UTS of the material).
  • Therefore, calibration tests were performed. In these tests, a strain gage was attached to the miniature specimen. Then, the specimen was load, where the external forces were incrementally increased, and the strain and the reaction forces of both load cells were measured (blue series, Fig 13), confirming the local plasticity.
  • After, a numerical model, encompassing plasticity effects, was developed, and some parameters (elasticity model of fixturing system, friction coefficient, bolt pre-load, etc) were “tuned” (small variation around an initial value), in order to provide a good agreement with experimental data (Fig 13, red series).
  • After, the experimental reaction force from the fatigue tests were used to calculate the elastoplastic stresses (black series, Fig. 14). It is important to emphasize that, at that time, the S-N results for standard axial specimens were not available yet.
  • Next, the standard specimens (axial) were tested (Fig 14, red series).
  • At that time, we did not expect such a strong agreement between the results of miniature specimens and axial ones. As the reviewer stated, due to the size effect, a superior fatigue performance of miniature specimens would be expected. The same effect was expected according to the type of tests (again, miniature specimens were expected to have superior fatigue performance, once they were subjected to bending, in comparison with standard ones, which were subjected to axial loads). But as explained before the higher cross section of the miniature specimens and disturbance to the linear distributed stresses in the cross section due to local plasticity would disturb the expect size effect result as shown by Nicoletto for fully elastic test and same cross sections of the axial and bending miniature specimens.

2) The authors developed a numerical analysis in order to correct the bending fatigue results for plasticity. In my opinion, the strain-hardening equation (plasticity model) used in non-linear FEM analysis should be explicitly presented in paper as well as any other assumption necessary to calibrate the numerical model, that is, superpose the FEM results to the experimental strain measurements.

A: Thanks for the feedback. The manuscript was reviewed and the relevant information regarding to the numerical model was added. Regarding the information added to the numerical model, parameters like bolt diameter and pre-load, friction coefficient and elasticity modulus of both specimen and gripping system were included, as well as the plasticity parameters for the specimen material.

3) The authors claim that the plasticity model used was based on the experimental tensile properties. Since there may be differences between the monotonic and cyclic stress-strain curves for a material, some additional discussion on the cyclic behavior of DED-processed Inconel 625 should be included in paper.

A: This observation was very relevant. It is true that the material is loaded above its yield stress (in most of the fatigue tests), which will lead to plastic deformation. However, this happens only on the first loading cycle. Since the load ratio, R, is bigger than 0, the material will work on an linear-elastic regime on the remaining cycles (which would not occur on a R=-1, for example).

An example is presented on the figure below. For that case, a maximum stress of 700 MPa was used, and the R=0.1 (as in the experimental fatigue tests). During the first load cycle, the material will be subjected to plastic deformation (cycle 1, loading). After, it will be unloaded (minimum stress, 70 MPa). On the successive cycles, the material will be loaded again to 700 MPa. However, on this case, it will no longer undergo plastic deformation; it will be “walking” over the grey curve (linear, from 700 MPa to 70 MPa).

The manuscript was updated, and a discussion regarding this topic was added.

This reviewer also points out some minor corrections to be done:   

- line 132: “… seen in Figure…”

A: Done

- line 150: “… half of the height…”

A: Done

- line 243: “… achieved, …”

A:Done

- line 267: Please explain what is the G-code.

A: Indeed the “G-code” was not properly explained. A short explanation was added, stating that g-code contains the information regarding the laser trajectory (deposition head) and process parameters (laser power, scanning speed, shield gas flow rate, powder feed rate, carried gas flow rate, etc.)

- Figure 11: Please consider to write “Logarithmic strain” instead “True strain”.

A: Done

- Table 2 and Table 3: please use “Yield Stress” instead of “Yield Strenght”.

A: The terminology used in the paper was according ASTM E8 standard which uses the yield strength wording. Anyway the suggestion was accomplished.

- line 302: “… and the yield and UTS were the under the…”? I did not understand this statement.

A: This sentence was reviewed as “The investigated material presented a yield stress similar to the wrought material, but lower than the other AMed materials. In addition, the studied material demonstrated the highest UTS among the four considered cases, leading to the highest strain hardening. On the other hand, the hardness was the lowest and the elongation was lower than the wrought material but well above the other AMed materials.”

- line 365: please write “axial fatigue specimen” instead of “tensile specimen”.

A: Done

Reviewer 2 Report

The manuscript represents a report on fatigue assessment of Inconel 625 alloy produced by DED from miniaturized specimens. The authors present lots of experimental results, and some of them are interest to the readers.

In the reviewer's opinion, there are some modifications should be made before the manuscript can be accepted for publication.
(1) the technical terms should be unified, such as DED or AM, Inconel 625 or IN625.

(2) the picture quality should be improved. For example, Figure 7, Figure 13…

(3) some of the results are marked by arrows in Figure 14 and Figure 15, but the author did not further explain why they do this.  

(4) the inverse slope (m) and stress range for 2 million cycles (Δσc) were summarized in table 3. Could the authors further explain what causes the difference?

(5) what causes the presence or absence of the fatigue striations in Figure 17?

Author Response

The manuscript represents a report on fatigue assessment of Inconel 625 alloy produced by DED from miniaturized specimens. The authors present lots of experimental results, and some of them are interest to the readers.

In the reviewer's opinion, there are some modifications should be made before the manuscript can be accepted for publication.(1) the technical terms should be unified, such as DED or AM, Inconel 625 or IN625.

A: Thanks for the feedback. All terms related to “IN625” were unified. The same thing was done to DED, AM was only used where a more general notation was required (referring to more than one process, like DED, LPBF, Solid State, etc.)

(2) the picture quality should be improved. For example, Figure 7, Figure 13…

A: The quality of the pictures were improved. Vector graphics were used in possible (charts, from example).

(3) some of the results are marked by arrows in Figure 14 and Figure 15, but the author did not further explain why they do this.  

A: This is the common way to refer the run-outs in S-N plots. An explanation was added to the manuscript, stating that series marked with arrows represent run-out tests (specimens that did not present failure after the specified number of cycles).

(4) the inverse slope (m) and stress range for 2 million cycles (Δσc) were summarized in table 3. Could the authors further explain what causes the difference?

A: A discussion regarding the causes of inverse slope and (Δσc) was added. The inverse slope m could reflect the impact of crack initiation and crack propagation phases. When crack propagation dominates, the inverse slope should present low values (m~3-4). When the fatigue crack initiation is the dominant damage regime (which is the present case), inverse slope m should be higher and sensitive to many parameters.

(5) what causes the presence or absence of the fatigue striations in Figure 17?

A: Thanks for the question. The discussion in the paper about the fractographic analysis was enlarged and improved since the third reviewer has claimed for it. However, just for reviewer information, the striations are an indicative of how much a crack grows when subjected to one cycle. If the crack growth rate is so small, the striations are not visible (they are very close to each other). This could occur when the fatigue crack propagation regime is in the near threshold regimes (see region 1 in the figure below). Also the striations observation depends on the complex crack-microstructures interactions.

[Image Ref]: A. L. L. da Silva, Advanced Methodologies for the Fatigue Analysis of Representative Details of Metallic Bridges. PhD thesis, Faculdade de Engenharia da Universidade do Porto, 2015.

In addition, the striation marks are easier detected when the crack propagates on a constant direction (orthogonal to the loading direction), as is less influenced by the material microstructure. For the analysed samples, the crack initiation region presented a Crystallographic Growth Region (CGR), this area was characterized by a very uneven surface, where the crack propagation did not follow the global direction of maximum stress. Instead, the microstructure was the dominant factor for fatigue, where favourable crystallographic planes led the crack propagation.

That is why, for both miniature and standard specimens, the striation marks were visible near the middle of the section, where the crack growth rates were big enough, and followed a nearly constant propagation direction (small changes in direction occurred due to pores and other microstructure features. The striations can be seen in Fig 17 c, Fig 18b and Fig 19 b (Fib 17 b and Fig 18 b on the previous version of manuscript).

In addition, for the miniature specimens, since the tests were performed under displacement control, when the crack grows, the cross sectional area of the specimen was reduced, leading to a reduction of the reaction (and external forces), for a constant displacement amplitude. Therefore, at the end of the fatigue tests, the stresses and crack growth ratios were small, and the striation marks were no longer visible. This part of the test were excluded from the analysis.

At the present stage, only a qualitative analysis of crack growth rate is presented, and terms like “small” or “big” crack growth rates were used. In the near future, crack propagation tests will be performed (the CT samples were already printed and machined, and are waiting to be tested). This will make it possible to know at which crack growth rate values (da/dN) the striation marks are visible or not.

Reviewer 3 Report

The article presents an interesting approach to the problem of high-cycle fatigue assessment of additive manufactured samples from Inconel 625 material. Still, the presentation of the results contains imperfections in both the visualization and the interpretation of the results. Most doubts were raised by the hardness test results and the fractographic analysis, and the graphic part that requires correction. For these reasons, significant changes should be made before publication.

The comments were introduced to the attached text of the article.

Author Response

The direction of tensile loads in relation to the direction of the layers is different for the two types of samples. Please explain the possibility of comparing the results.

A: The first image (1 a) was not clear enough, therefore it was modified. Now the comparison between miniature specimens and tensile ones are more straightforward. Both have their loading direction parallel to y-axis.

Please explain the possibility of comparing the results of the fatigue tests for other control modes (force / stress, displacement). Why the same control method was not used?

A: Indeed, ideally both tests would be performed under the same control method. However, during the tests of miniature specimens, the only available machine was the one with displacement control. Also the concept behind the miniature specimens is to allow cost effective fatigue testing. In this sense, the usage of electromechanical testing machines instead of servo hydraulic ones would allow also to contribute to the cost efficiency. It is possible to compare methods, under certain conditions. During the crack nucleation stage, the specimens have a constant cross section (since there is no significant reduction of the section caused by cracks). Therefore, for the miniature bending specimens, during this stage, since the stiffness of the system is constant (and, as always, the displacement range), the reaction forces will be constant as well. Consequently, it is safe to say that, prior to any macroscale cracks are present (which would reduce the cross section of the specimens), it is possible to compare directly the force and displacement control. In other words, at this stage, the displacement control will present a similar behaviour to the force control method. A more complete explanation about the possibility of comparing both results were added to the manuscript.

Please describe more details about the strain measurements, as the classical measurement methods with a gauge length of 25 mm allow obtaining a strain of at least 2.5%.

A: The same strain gage (MMF307385) was used for both miniature and tensile specimens. Since the miniature specimen (“Nicoletto”) was very small (5 mm wide), it was not possible to use a strain gage much bigger than that. However, for the tensile specimens, the strains were also measured using Digital Image Correlation, which were able to measure larger strains (for the present case, strains near 40% at failure). It would be possible to use a clip gauge (on the opposite face of the DIC one). However, a strain gage (for small displacements) was able to properly characterize the elastic regime (elasticity modulus) and the DIC was able to properly measure both elastic and plastic zones. When the review refers a gauge length of 25 mm we believe he is referring the clip gauge displacement transducers which were not considered in this study due to uncertainties introduced due to the contact issues.

Please describe why such a distance between successive hardness measurements was used and the validity of the base plate measurements. More details on the impact of successive layers could be obtained based on the microhardness

A: The initial idea was only to provide an average value of hardness on the sample. Nevertheless the idea of the reviewer is very interesting and for sure accounted in future works to address the variability of properties across deposited layers.

Please explain why this area was considered in the hardness measurements?

A: The hardness measurements on baseplate and near base plate (around 2 or 3 mm) does not provide any valuable information for the present work discussion, since these regions were not included at the final specimens material volume (tensile and fatigue ones), due to the separation process. However, this information might be relevant for other MAM topics, for example the DED repair applications, where both the baseplate (or a part to be repaired, which acts analogous to a baseplate) and the first layers mechanical properties are very important. It would be possible to present only the hardness of the regions presented on the final part (z position above 3 mm). However, the time required for performing 5 (only final part) or 9 (baseplate and part) hardness measurements would be nearly the same.

Please correct the description in Figure 10b as it is not legible.

A: Done

Please explain the relationship between the base plate hardness and the cooling rate (validity of the hardness measurements)

A: As said before, this information about the base plate included in the manuscript it is a marginal information in the manuscript, which according to the authors could be usefull for some readers but should be not given to much attention because interface problems are out of the scope of this research paper. Nevertheless we would like to reply the reviewer in this reply document.

The baseplate was a steel part (40CrMnMo7). This alloy is susceptible to quenching. Since the baseplate regions near the interface (first layer deposition) are subjected to high temperatures and high cooling rates, these regions can experience improvement in its hardness due to the quenching process (presence of martensitic microstructure). Farther from these interface (away from the heat affect zone), the baseplate microstructure does not experience temperatures above the transformation ones, therefore they will present lower hardness. Additionally, in order to present good mechanical properties, it is essential that every deposited layer presented a strong bonding with the previous one (or the baseplate, in case of the first deposited layer). This is achieved by a “penetration” of the deposited layer on the previous one. Therefore, when performing measurements very close to the interface (z=0 mm), we are not measuring the baseplate hardness itself, but an alloy, containing a mixture of the baseplate material (steel) and the deposited one (Inconel 625). As stated before, the baseplate values were only provided as an additional information, however the effect of the material deposition on the baseplate properties is not the objective of the present work.

Please describe why the true stresses were compared, not the engineering stresses, in my opinion, it is shown differently in the quoted literature.

A: The manuscript was updated with the proper stress (engineering). We checked again the quoted literature, and indeed the UTS values presented are the engineering ones.

As shown in figure 11, there is an elongation deviation. Please describe why not included in table 3?

A: The description was not clear enough. The “deviation” is a comparison between the present work and the literature review [ref 11]. When it is said that the Deviation of the present work is “-“, the original goal was to estate that there is no reason to present this value, once it is a comparison between the present work with itself. The description was updated, in order to clarify this question. Also, the elongation (and all presented data on table 3) was an average of the measure values.

Please explain which number of samples is correct?

A: Thanks for the note, the total number of samples was wrong and updated now in the revised manuscript.

Please explain how the stress adjustments were made to make the comparison? There are no formulas in the content of the article that would allow for such a correction.

A: The equation for stress correction from R=0.1 to R=-1 was added to the manuscript. It is based on Smith-Watson-Toper mean stress correction (SWT), equation (4).

Please explain how the fatigue striations were found on the tensile samples? The text does not match the captions under Figure 17.

A: A misleading term was used, “Tensile sample”. The term was now updated to “standard axial fatigue sample”.

Please complete the article with an analysis of fracture surfaces, because in its current form, individual photos without an appropriate description cannot be treated as an analysis.

A: The discussion regarding fracture surface was improved.

Please explain what this description in the photo of the fatigue fracture means?

A: The discussion was improved, encompassing the description of each regions presented in  Fig 18.

The presented conclusions are too far-reaching assumptions, e.g. high-cycle fatigue tests are carried out in the load areas below the yield point, please compare the results are presented in Figure 16.

A: The manuscript was not clear enough regarding this issue. The values presented in Fig 16 were the equivalent stresses if the tests were conducted on R=-1 (according to SWT approach). This conclusion (about the specimens being tested above yield stress) is about the R=0.1 (Fig 15), which was the load ratio used on experimental tests for both standard and miniature specimens. The manuscript was reviewed, and a more detailed explanation about the ratio conversion (from R=0.1, Fig 15, to R=-1, Fig 16) was included.

Round 2

Reviewer 1 Report

The questions and points raised by this reviewer were fully addressed by the authors and the paper was correspondingly revised.

Reviewer 3 Report

Dear Authors,
thank you for taking into account my comments.
The article is now much clearer and can be published in its current form.